**DOI: 10.1038/ncomms15626**　　**OPEN**

# Attosecond interferometry with self-amplified spontaneous emission of a free-electron laser

Sergey Usenko[1,2], Andreas Przystawik[1], Markus Alexander Jakob[2,3], Leslie Lamberto Lazzarino[3], Günter Brenner[1], Sven Toleikis[1], Christian Haunhorst[4], Detlef Kip[4] & Tim Laarmann[1,2]

Light-phase-sensitive techniques, such as coherent multidimensional spectroscopy, are well-established in a broad spectral range, already spanning from radio-frequencies in nuclear magnetic resonance spectroscopy to visible and ultraviolet wavelengths in nonlinear optics with table-top lasers. In these cases, the ability to tailor the phases of electromagnetic waves with high precision is essential. Here we achieve phase control of extreme-ultraviolet pulses from a free-electron laser (FEL) on the attosecond timescale in a Michelson-type all-reflective interferometric autocorrelator. By varying the relative phase of the generated pulse replicas with sub-cycle precision we observe the field interference, that is, the light-wave oscillation with a period of 129 as. The successful transfer of a powerful optical method towards short-wavelength FEL science and technology paves the way towards utilization of advanced nonlinear methodologies even at partially coherent soft X-ray FEL sources that rely on self-amplified spontaneous emission.

[1] Deutsches Elektronen-Synchrotron DESY, Notkestraße 85, Hamburg 22607, Germany. [2] The Hamburg Centre for Ultrafast Imaging CUI, Luruper Chaussee 149, Hamburg 22761, Germany. [3] Department of Physics, University of Hamburg, Hamburg 22761, Germany. [4] Faculty of Electrical Engineering, Helmut Schmidt University, Hamburg 22043, Germany. Correspondence and requests for materials should be addressed to T.L. (email: tim.laarmann@desy.de).

Science with short-wavelength free-electron lasers (FELs) has enabled multiple breakthroughs covering the broad range from basic research in life sciences to applications in material science and catalysis. Particularly, the high degree of spatial coherence of the light field allows for key applications, such as serial-femtosecond X-ray crystallography, using the well-established and robust self-amplified spontaneous emission (SASE) of FELs[1]. Recently, temporal coherence provided by seeded FELs moved into the focus of interest[2–4]. It has been shown that full control over the light phase allows for a new class of light-phase sensitive experiments in the short-wavelength limit[5–7], such as nonlinear four-wave mixing[8] and attosecond ($1\,as = 10^{-18}\,s$) coherent control[9]. These X-ray methodologies give a new twist to modern laser science striving to unravel energy, charge and information transport phenomena on attosecond time and nanometre length scales in matter, materials and building blocks of nature[10]. At the heart of any atomistic understanding of complex functionality such as chemical bond formation is the ultrafast motion of electrons, which is dictated by the laws of quantum mechanics. Particularly, information on the phase evolution of a wave packet, which describes electronic and/or nuclear structural changes, holds the key for its control in space and time. In optics, direct phase information of the propagating light wave is derived by interferometric techniques[11]. In the last decade analogous methodologies have been developed to derive similar information on ultrafast electron wave packet dynamics in atoms, molecules, clusters and solid state materials. The workhorse in these studies are extreme-ultraviolet pulse trains provided by high-harmonic generation (HHG) sources with pulse durations of a few 100 as and even below separated by a half-optical cycle of the femtosecond drive laser[12,13]. Meanwhile, single attosecond pulses can be generated[14–17] or spatially isolated from the attosecond burst by ultrafast wavefront rotation[18]. In two-colour attosecond electron wave packet interferometry typically the extreme-ultraviolet pulse sequence ionizes the sample and a fraction of the phase-locked near-infrared drive laser induces a momentum shear between subsequent outgoing electron wave packets[19]. The resulting interferograms in momentum space allow to reconstruct molecular orbitals[20] or to trace most fundamental processes such as photoionization of an electron from its parent atom or molecule in real time. In analogy, time-domain interferometry between the extreme-ultraviolet bursts originating from consecutive laser half-cycles in the HHG process, where the atomic potential barrier is modulated on optical sub-cycle timescales, allows to watch the electronic motion pictures during strong-field ionization and electron recollision in a time window of 200 as (ref. 21).

Prerequisites for advanced nonlinear phase-sensitive extreme-ultraviolet and soft X-ray experiments are phase stability of the light source, and the ability to control the temporal (longitudinal) phase of the light wave on the corresponding sub-cycle attosecond time scale. Phase-coherent pulse synthesis, metrology and application using table-top laser systems have been pushed to the limits and beyond by the HHG community. However, nonlinear absorption cross sections, as well as the HHG conversion efficiency decrease with increasing photon energy. Seeded FELs promise to overcome these limitations. The accelerator-driven light sources provide sufficiently intense femtosecond pulses with well-defined polarization and a high degree of coherence, both transverse and longitudinal[2,3]. With an FEL peak power in the Gigawatt range, nonlinear processes can be induced efficiently at significantly shorter wavelengths. It has been demonstrated that two-colour lasing from the seeded FERMI FEL tuned to the first and second harmonic ($\omega + 2\omega$)

of the FEL resonance frequency is phase stable to each other, allowing for coherent control experiments with a temporal resolution of 3 as (ref. 9).

Compared with seeded FELs, the longitudinal coherence of SASE FELs is poor[22,23]. Their spectral distribution fluctuates from shot-to-shot and comprises a series of uncorrelated, coherent spikes (longitudinal modes) within the amplification bandwidth of the FEL. However, it was demonstrated that also partially coherent FEL beams can be characterized in time domain down to the attosecond timescale using a variant of frequency-resolved optical gating[24], which is a well-established spectrographic autocorrelation technique for complete pulse reconstruction[25]. Recently, measurements of second- and higher-order intensity correlation functions by means of Hanbury Brown–Twiss interferometry provided quantitative information on longitudinal coherence properties of SASE pulses[23]. A detailed analysis of the experimental data taken at an FEL wavelength of 5.5 nm and evaluated in the framework of statistical optics gives a degeneracy parameter on the order of $10^9$. In other words, an individual SASE spike selected by a grating monochromator comprises up to $10^9$ photons within the coherence time of a few femtoseconds, which is similar to intense optical laser pulses and within reach of table-top HHG sources[26].

In the following we demonstrate attosecond phase control of extreme-ultraviolet light waves by generating two phase-locked replicas of SASE FEL pulses and observing their interference directly in the time domain. This opens the door for time-domain interferometry enhancing the information content of nonlinear optics experiments in the soft X-ray range even at partially coherent SASE FEL sources.

## Results

**FEL source.** The present study has been performed at the soft X-ray FEL in Hamburg, FLASH[27]. Its fixed-gap undulator line, called FLASH1, is a dedicated SASE FEL. The electron bunch charge was 0.32 nC and the electron energy of 410 MeV results in a photon wavelength of 38 nm. The average pulse energy was about 78 µJ, which corresponds to roughly $1.5 \times 10^{13}$ photons per pulse at this wavelength. The measurements were performed at the PG2 monochromator beamline of FLASH1 (refs 28,29). The beamline comprises a plane grating (200 lines per mm), collimating and focusing mirrors. The exit slit with variable width was used to select a single SASE mode. The monochromator was tuned to the first diffraction order resulting in dispersion of $65.2\,meV\,mm^{-1}$ at 32.63 eV photon energy in the exit slit plane. The slit width was set to 70 µm, which corresponds to an FEL bandwidth of 4.6 meV (0.005 nm) and a total transmission of $1.1 \times 10^{-3}$. A characteristic single-shot SASE FEL spectrum recorded during the experiments is shown in Fig. 1 together with an average spectrum of 1,800 pulses. The FLASH pulse exhibits many coherent longitudinal modes. It is important to note that the selected spectral bandwidth is significantly smaller than the width of individual modes. Thus, the radiation field comprising approximately $10^{10}$ FEL photons per pulse passing the exit slit possesses a high degree of spatial and temporal coherence.

**Experiment and data processing.** The experimental setup is depicted in Fig. 2, and described in more detail in the Methods section and Supplementary Notes 1–3. A reflective split-and-delay unit (SDU) plays the key role for phase-resolved one-colour pump-probe spectroscopy at this wavelength. The SDU splits the wavefront of the incoming FEL pulse uniformly across the beam profile by two interleaved gratings and provides two pulse replicas with a variable delay. The two gratings generate

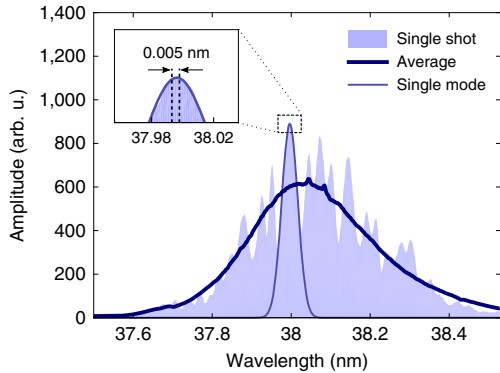

**Figure 1 | Spectral distribution of SASE FEL pulses.** A typical single-shot spectrum of self-amplified spontaneous emission (SASE) and the average spectrum of 1,800 pulses are shown. The free-electron laser (FEL) spectral bandwidth selected by the 70 μm wide monochromator exit slit is indicated in the inset. Individual longitudinal FEL modes have a significantly broader spectrum exemplified by the Gaussian fit, that is, the transmitted photon flux exhibits a high degree of longitudinal (temporal) coherence.

a number of diffraction orders and in each order the wavefronts of the two partial beams are parallel. We like to emphasise that in contrast to a conventional half-mirror SDU, the above geometry provides collinear propagation of both pulse replicas and thus constant phase difference across the beam profile. This enables to record phase-resolved autocorrelation signals with maximum contrast[30]. Soft X-ray interferometry requires high surface quality and position control of the reflective optics on the sub-wavelength, that is, nm length scale. Slope errors result in wavefront distortion of the corresponding partial beam and reduce the mutual spatial coherence of the pulses to the order of several per cent. However, even this value is sufficient to observe rich interference contrast as a function of attosecond time delay between the pair of FEL pulses, as it will be detailed below. A spherical mirror focused the pulse replicas into Xe gas. The generated ions were recorded with a time-of-flight spectrometer (see Supplementary Note 2 for details). The spectrometer was operated in a mass-sensitive spatial imaging mode, that is, $Xe^+$ ions were projected onto a position sensitive detector. The experiment is capable of single-shot data acquisition and single-shot characterization of relative-phase differences and pulse intensities, thus the 100% intensity fluctuations behind the monochromator are not a problem. It goes without saying that in linear spectroscopy data averaging performed over many pulses does not present any difficulties. As far as non-linear spectroscopy is concerned, the intensity dependence of the systems response is measured simultaneously by sorting the data according to FEL pulse intensity. The single-shot ion images were taken synchronized with the FEL pulses at the 10 Hz repetition rate of the accelerator. A 500 ns wide temporal gate set on the ion detector was tuned to the arrival of $Xe^+$ ions, which have a time-of-flight of 9.5 μs. The exact autocorrelation delay for each pair of FEL pulse replicas was derived by simultaneously imaging the surface topography of the split-and-delay unit with nm-precision using an in-vacuum white-light interferometer. For data processing the ion images were sorted according to the actual delay. The distribution was binned into 15.84 as (1/8 of the optical cycle at 38 nm) time slots, which can be regarded as the effective step size of the present experiment sampling a total range of 450 as. The images within each time bin were summed-up and normalized to the number of shots. The ion yield distribution across the focus was obtained by integrating the normalized images along the dimension corresponding to the beam

propagation (Fig. 3a). Because of the constant relative phase of the coherent FEL pulse replicas for each delay, the amplitude ratio between neighbouring diffraction orders changes as a function of pulse separation. The resulting fringe contrast is clearly visible in Fig. 3b. It monitors the field interference when varying the relative phase within the optical cycle, that is, the coherent light wave oscillation with a period of $129 \pm 4$ attoseconds at the FEL wavelength of 38 nm.

## Discussion

Our study opens up the door for high-contrast time-domain interferometry in nonlinear phase-sensitive spectroscopic studies even at partially coherent SASE sources. Full control over the relative temporal phase in FEL pulse replicas provides opportunities to trace energy and charge migration in systems of increasing complexity with unprecedented spatial and temporal resolution. It makes the local electronic structure and dynamics accessible, that is, controllable. In the following we give some perspectives of extreme-ultraviolet attosecond interferometry applications at FLASH. For example, a hot topic in modern attosecond science is to follow the birth, migration and fate of an electronic wave packet that is a superposition of cationic eigenstates on ionization of large but finite quantum systems[31]. Advanced applications of this kind become possible by using relatively low electron bunch charges of 20–60 pC in the FLASH machine, while keeping the electron peak current high. It results in only 1–2 longitudinal modes per FEL pulse on the 10 μJ level and a high degree of longitudinal coherence without the need for a monochromator. These pulses are sufficiently short (only a few fs)[32], that is, broad bandwidth to coherently couple several cationic eigenstates forming an electronic wave packed. Its ultrafast propagation is expected to show amplitude oscillations of the probability density distribution in space and time followed by subsequent charge localization[33,34]. The light-induced dynamics is triggered by coherent coupling of many-body states mediated by electron correlation. The transformation of electronic orbitals, that is, the coupling between different electronic configurations proceeds on a timescale that is short compared with the characteristic timescale of nuclear motion[35]. Pioneering experiments were performed by Weinkauf and coworkers on site-selective reactivity leading to molecular fragmentation[36–38]. We note, that the M- and K-shell resonances of C, O and N lie within the tuning range of modern FELs, for example, FLASH, FERMI and LCLS, respectively. Thus, element-specific studies on organic compounds with impact in chemistry, biology and life science are within reach after we extend our method down to the few-nm wavelength range. Thicker substrates are less prone to warping due to internal stress. Together with optimized processing procedures this will improve surface planarity significantly. In addition, at shorter excitation wavelengths we will reduce the incidence angle on the gratings in the setup (currently 22°) to maintain sufficient reflectivity. This has a positive effect on the phase resolution, as the actual optical path length difference between the two copies of the FEL pulse is given by $2d \sin\alpha$, with α being the angle to the surface and $d$ being the relative displacement of the surfaces. That means halving the incidence angle, while also halving the excitation wavelength, keeps the relative phase resolution constant. Therefore, the approach is scalable to the few-nm excitation wavelength (a more detailed discussion is given in Supplementary Note 1). We like to point out that phase-shifters between undulator modules of seeded FELs allow at maximum a delay of a few hundred attoseconds[39], while twin-pulse seeding schemes are limited to clearly separated pulses to avoid interference between

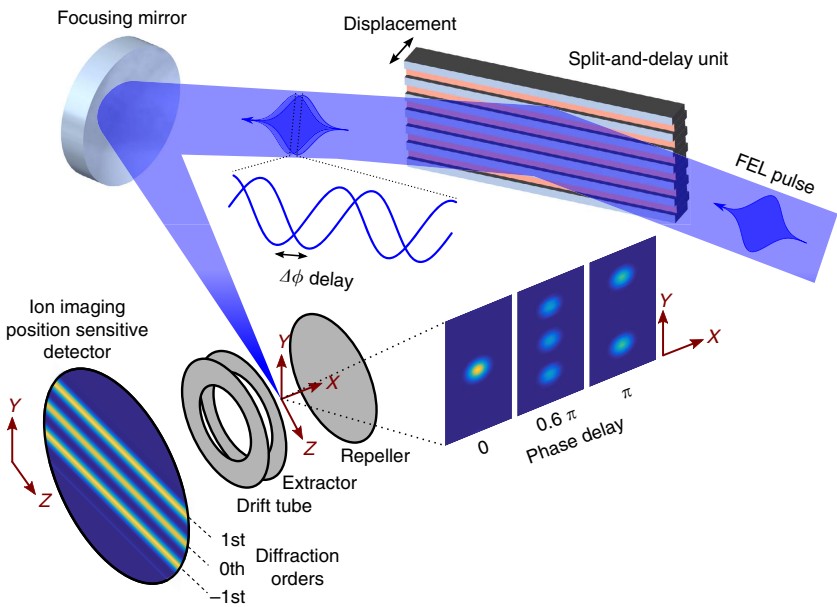

**Figure 2 | Experimental setup.** A free-electron laser (FEL) pulse with a central wavelength of 38 nm is diffracted from the split-and-delay unit comprised of two interleaved gratings, each with a 250 μm period. The generated 'double-pulse' wavefront is then focused with a spherical mirror ($f = 300$ mm) resulting in several diffraction orders separated by 46 μm. The spatial FEL irradiance distribution depends on the temporal separation of the pulse replicas and generates $Xe^+$ in the centre of a time-of-flight spectrometer. It comprises a set of electrodes used for ion extraction and focusing, a drift tube and a position-sensitive detector (PSD), where the electron output from a micro-channel plate impacts a phosphor screen. The fluorescence signal from the PSD is imaged with a CCD camera. The achieved spatial resolution in the ionization volume of 4.6 μm is sufficient to resolve the different diffraction orders.

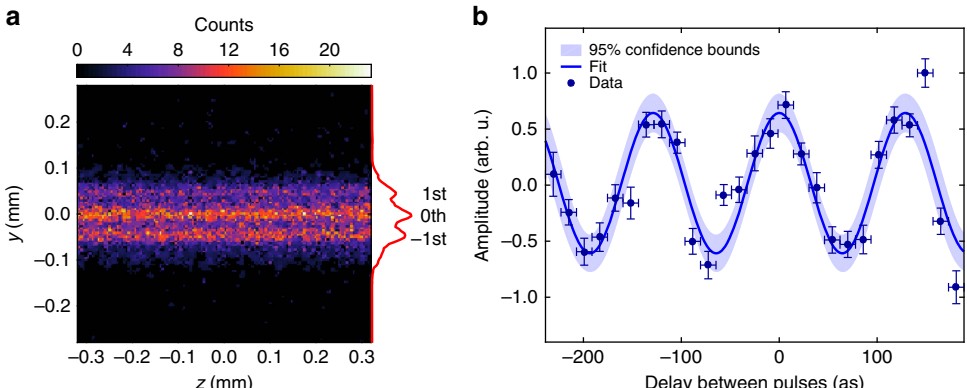

**Figure 3 | Interferometric autocorrelation.** (**a**) Spatial image of the ionization volume reflecting the irradiance distribution along the free-electron laser (FEL) beam path. It is derived by accumulating single-shot ion images from 3,000 pulse pairs with individual relative-phase delays covering 450 as. The projection along the FEL beam direction clearly shows different diffraction orders. (**b**) The FEL light wave oscillation is monitored by plotting the normalized ratio between 0th and 1st diffraction order as a function of relative phase delay of each pulse pair. The horizontal error bars denote the bin width of 15.84 as. Vertically, $2\sigma$ error bars are given for the fitted ratio between the orders. The measured optical cycle at the FEL wavelength of 38 nm is $129 \pm 4$ as.

the laser seeds, imposing a minimum delay[40]. In contrast, our present setup can control the relative delay between two extreme-ultraviolet excitation pulses from $-50$ fs up to $+574$ fs with attosecond precision continuously and is not tied to a particular photon source.

In case of glycine, extreme-ultraviolet interferometry can be applied to the inner valence region. Here, the cationic states form a quasicontinuum of close lying energy levels[41]. The ensuing electron dynamics in this model system is a current research focus in modern attosecond science, as glycine is the simplest of the twenty natural amino acids acting as molecular building blocks of more complex biochemical compounds such as peptides and proteins.

Another interesting proof-of-principle experiment using extreme-ultraviolet attosecond interferometry, as a tool to study and possibly control the correlated motion of electrons and holes is to follow the Auger-decay in a $CH_3I$ molecule. It is a prototypical polyatomic quantum system whose photophysical and photochemical properties including extreme-ultraviolet-induced ionization of inner-shell electrons are well-studied[42–44]. At a photon excitation energy of 68.6 eV (18.1 nm) the valence and inner-shell photoionization cross-sections are similar[44]. The removal of a valence electron from the $CH_3I$ molecule results predominantly in a singly charged $CH_3I^+$ ion, whereas the generation of an inner-shell $I(4d)$ hole leads to a doubly or triply charged final state after subsequent relaxation via Auger decay.

The Auger decay of inner-shell vacancies already involves the interplay of at least three electrons, and cascaded Auger decay may couple even more. Most of the created multiply charged cations then break up into charged fragments[45]. In total, several electrons are excited into the continuum, where a comprehensive time-dependent study of their yield, energy and angular distribution will reveal detailed information on amplitude and phase of participating partial electron waves. Fundamental questions can be addressed such as: what is the decoherence time of an excited correlated state in an atom? Can this be manipulated (controlled)? What is the degree of controllability in the systems' response, that is, how does it depend on the relative phase separation of pulse replicas upon resonant excitation? Is the coherence of an excited correlated state preserved on coupling to nuclear degrees of freedom in a molecular system during its fragmentation?

Similar questions at the heart of many-body quantum physics emerge in extreme-ultraviolet-excited $C_{60}$ fullerenes[46], which can be answered by means of attosecond interferometry in combination with angular-resolved photoelectron spectroscopy. Here, single-mode SASE pulse replicas drive plasmons in $C_{60}$ that produce giant resonances in its photoabsorption spectrum at about 20 and 40 eV, respectively. The energetic pulses provide on the order of $10^{13}$ coherent photons allowing for efficient excitation of the underlying electronic state manifold. Of particular interest is the coherent excitation in the spectral overlap region between the so-called surface- and volume-plasmon resonances, that is, at 30 eV photon energy[47]. Complex multi-electron dynamics is induced[46], which results in multiply charged interaction products. A detailed phase-sensitive analysis of the emitted electron wave packet properties holds the key to unravel the basic mechanism and the relevant time scales. Because of the large charge conjugation, its finite energy gap, and quantum confinement of electronic states, $C_{60}$ may be viewed as an interesting intermediate case between a condensed matter system and a molecule[48]. One has to keep in mind that molecular bound–bound transitions between $\pi \rightarrow \delta$, $\sigma \rightarrow \pi$, $\pi \rightarrow \sigma$, and $\sigma \rightarrow \delta$ orbitals contribute to photoabsorption in the energy range between 5 and 30 eV[49].

Last but not least, a well studied many-body quantum system that exhibits a collective electronic response to extreme-ultraviolet irradiation known as the $4d$ giant dipole resonance is the xenon atom. Recently, it has been shown that the excitation comprises two fundamental collective resonances[50]. According to theoretical work[51], close-lying electronic states split into two far-separated resonances through electron correlation (configuration interaction) involving the $4d$ electrons. In this example, extreme-ultraviolet attosecond interferometry can be used for a deeper understanding of the dynamic response of the underlying resonances. Phase-coherent, pulse replicas with pulse durations of a few fs generated from the single-mode FEL at a central wavelength of $\sim 13.5$ nm excite the electronic state manifold. The FLASH light wave oscillation would be as fast as $\sim 40$ as driving the multielectron excitation into the continuum. Thus, the emitted electrons that are detected and characterized as a function of relative phase delay carry valuable information on amplitude and phase of participating partial waves forming the actual outgoing electron wave packet. It is important to stress that the low-bunch charge SASE operation (single-mode) will not require any monochromatisation for attosecond extreme-ultraviolet FEL interferometry.

## Methods

**Split-and-delay unit.** The split-and-delay unit (SDU) employed in the experiment consists of two gratings of different but complementary design. Both gratings are produced from high-quality silicon wafers. The first grating is manufactured from a $60 \times 35 \times 1$ mm$^3$ wafer. The central $20 \times 10$ mm$^2$ area is processed with a circular saw to form a slotted grid (Supplementary Fig. 1a). The grid consists of 150 μm wide, 10 mm long slits with a period of 250 μm. The second grating is a ridged structure designed to fit into the slits of the grid (Supplementary Fig. 1b). Its reflective face presents a pattern of 8 mm long 100 μm wide ridges protruding from the substrate for 1.25 mm. When interleaved the two gratings form a sequence of 100 μm wide reflective facets with neighbouring elements separated by 25 μm gaps belonging to different gratings as shown in Supplementary Fig. 1c. The surface quality of the gratings was characterized using a home-built white light interferometer. A heightmap of the ridged grating for the FEL illuminated area is shown in Supplementary Fig. 2a. The surface shows a r.m.s.d. of 3.2 nm from a perfect plane.

After reflection from the grating assembly the FEL beam is focused by a spherical mirror into a xenon gas sample. Diffraction orders result in a sequence of focus replicas distributed along the dispersion direction in the focal plane of the mirror. The irradiance distribution of the FEL beam diffracted from the ridged grating and focused by a mirror with a 300 mm focal length is shown in Supplementary Fig. 3. The image displays several beam replicas (including orders up to fourth) separated by 46 μm. In general, the intensity distribution between different orders (envelope) for a given wavelength depends on the fill factor of the used gratings, which is 0.8 for the assembled device. Therefore, the intensity of high orders drops rather quickly and only the three brightest ones ( − 1st, 0th, 1st) are clearly visible. When the two gratings are interleaved the relative intensity of odd and even orders becomes a function of the relative displacement of the gratings as illustrated in Fig. 2. In principle, any diffraction order can be chosen to observe the time-dependent signal. The zeroth order is typically the most convenient to work with due to its highest intensity and thus the best signal to noise ratio.

Both gratings of the SDU are mounted on a specially designed mount. The mount has three piezo actuators: two used for parallel alignment of the gratings and one used to control the relative displacement of the gratings along the surface normal (for example, delay between the pulse replicas). The translation stage used for delay generation provides a travel range of 250 μm. The incident FEL beam has a 22° angle of incidence to the surface of the gratings. In this geometry the present actuator allows to generate delays in a range − 50 to 574 fs.

**Single-shot delay diagnostics.** In the experiment the surface profile of the SDU is monitored for each FEL pulse by a home-built Michelson white light interferometer (WLI). It includes a white light diode, a broad band beam splitting cube, a silicon reference mirror mounted on a piezo actuator, and a camera. All the components except the camera and the diode are placed in-vacuum in close proximity to the SDU. The camera images the central region of the SDU $10 \times 10$ mm$^2$ in size with a lateral resolution of 10 μm. The WLI images are recorded simultaneously with the ion data in synchronization with the FEL pulses at the machine repetition rate of 10 Hz. The exact displacement of the two gratings (and thus the delay) for each FEL pulse pair is derived from the appearance of the white light interference pattern. The system allows to determine the SDU displacement with a precision down to 3 nm, which translates into 7.5 as of delay in the present experimental geometry.

It is important to note that the WLI exposure time is significantly shorter ($\approx 200$ μs) as compared to the time scale of dominant vibrations in our experiment. Thus, single-shot WLI images give snapshots of the SDU profile for each FEL pulse.

**Data acquisition and analysis.** The ion and WLI images were recorded in synchronization for each FEL pulse. The single-shot data files were sorted according to the actual delay between the pulse replicas. The obtained distribution was binned into 15.84 as bins (1/8 of the optical cycle at 38 nm central wavelength). Typically, a few thousand spatial images of ions in each bin were then averaged.

**Data availability.** All relevant data are available from the corresponding author on reasonable request.

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

## Acknowledgements

We thank Hubertus Wabnitz, Ivan Vartaniants and Evgeny Saldin for fruitful discussions. This work was supported by the Deutsche Forschungsgemeinschaft through the excellence cluster 'The Hamburg Centre for Ultrafast Imaging (CUI)—Structure, Dynamics and Control of Matter at the Atomic Scale' (DFG-EXC1074), the collaborative research centre 'Light-induced Dynamics and Control of Correlated Quantum Systems' (SFB925), the GrK 1355 and by the Federal Ministry of Education and Research of Germany under Contract No. 05K16GU4.

## Author contributions

T.L. conceived the idea and coordinated the project. The experimental setup was designed by all authors. The ion spectrometer was developed by S.U. The split-and-delay unit was developed by S.U., A.P., L.L.L., C.H. and D.K. The white-light interferometer for online relative phase analysis was developed by S.U., L.L.L., M.A.J. and A.P. The experiment was performed by S.U., A.P., M.A.J., L.L.L., G.B., S.T. and T.L. The data were analysed by S.U. The results were interpreted by S.U. and T.L. The manuscript was written by T.L with contributions from S.U. and A.P., as well as input from all authors.

## Additional information

**Competing interests:** The authors declare no competing financial interests.

**Publisher's note**: 

