## [Peer Review File · Nature Communications]

Reviewers' comments:

Reviewer #1 (Remarks to the Author):

The manuscript presents the application of the interferometric autocorrelation technique on a Free Electron Laser SASE pulse by picking one single mode of the pulse's spectrum and splitting it into two replicas with a grating type split and delay mirror. The two replicas were focussed into a position sensitive ion time of flight spectrometer and delay dependent singly charged xenon ions signal was measured. The results demonstrate the successful adaptation of this technique to partially coherent pulses provided one can generate or select one single mode.

General remark:

The authors present a convincing technique to control the relative phase of two replicas of a coherent pulses on the attosecond time scale. They have taken into accounts remarks on the previous manuscript version by adding experiment perspectives and discussed in more details the characteristics and limits of the technique in the additional material. Nevertheless, the four experimental perspectives are discussed for as many different FEL conditions. As an example the measurements have been done for a wavelength of 38 nm while the requirement for the xenon giant dipole resonance probing requires 13.5 nm wavelength. Moreover, the authors discuss the possibility of probing charge dynamics in biomolecules with single FEL mode through low bunch charge operating conditions at FLASH. This, as showed in the response to referees, corresponds to 7 nm wavelength pulses, how good does the split and delay works at this wavelength? We also understand that in this particular case the dispositive should be used as a "regular" pump-probe delay stage where one take into account the overlap between the two pulses envelopes with a very large delay range (-50 to 574 fs). Part of this is discussed in the supplementary material but I think it should be discussed in the manuscript as it would demonstrate unambiguously the versatility of the dispositive.

To conclude, the authors present an interesting method to control the delay between two FEL pulses, allowing a phase control at the attosecond time scale, moreover they responded to the first manuscript comments and provided the additional informations required but I think they should make a real case of the capabilities and limits of the technique not only in the supplementary material but in the manuscript too. Provided those changes, I would recommend this work to be published in Nature communication.

Additional remarks:

Lines 166-170: This paragraph should appear before because it is one of the best argument of the technique for future applications. Also we would like a reference (there is one in the response to the referees.)

Lines 163-166: The site selective excitation requires wavelength in the nm range. This argument would benefit from a discussion on short wavelenght limits of the setup in the manuscript.

Line 165: LCLS too has the capability to cover the C, N and O K-edges.

Reviewer #3 (Remarks to the Author):

The manuscript reports experimental demonstration of generating two replica pulses with phase control between them from a SASE FEL source, at the FLASH FEL with wavelength of 38 nm. In the updated version, the split and delay unit, which plays an important role in the experiment, was added with detailed descriptions in supplementary material part. More discussions have been added about the potential applications with this developed technique. Overall, this is an interesting scheme, overcoming the disadvantages of the SASE pulses which have poor longitudinal coherence would expand SASE FEL applications in the cases where require accurate phase control.

I would recommend the publication of this manuscript in this journal.

Some remarks:

The scheme chose a very narrow bandwidth smaller than a single spectral spike which introduces large fluctuation on the intensity after the mono. For the low charge mode with only one or two spikes, the fluctuation problem is similar. The authors explained in the reply of the single-shot feature and data sorting analysis. This is fine for some experiments. I think it would be useful to point out this fluctuation issue in the main text and discuss briefly that the single-shot characterization feature would overcome this issue in data analysis.

Reviewer #4 (Remarks to the Author):

The paper proposes an experimental method to perform Michelson-type interferometry using special type of combo grating mirrors in XUV spectral range (wavelength 38 nm) with attosecond temporal precision (15.8 as). By itself, the paper is well write and the experimental data are clear and convincing. However, few innovative results are present inside the manuscript. Indeed the general idea to use a couple of gratings as split and delay mirror to perform Michelson-type interferometry is not new in EUV spectral range and has been already successfully proposed in a very similar publication by Gebert et al. *New J. Phys.* 16 07304 (2014). Moreover, the use of poor longitudinal coherence source like SASE-FEL source imposes the use of a monochromator to enhance the spectral purity of the radiation illuminating the splitting and delay unit, having strong penalty for the final photon flux at the sample plane, in the current experimental data about 80 nJ. The authors claim about the possibility to use single spike operation of SASE-FEL using low charge e-bunch. However, using different scheme of light generation, seed-FEL has already demonstrated the possibility to control the relative phase between two different pulses with better (3 as, K.Prince et al. *Nature Photon.* 10, 176-179 (2016)) or at least comparable performance (15 as, Gauthier, D. et al. *Phys. Rev. Lett.* 116, 024801 (2016)) with respect to mechanical based devices without restriction on photon outputs. This work would have benefited by a simple experimental data demonstrating with a non-linear process the capacities of the proposed unit. For those reasons, I considered this manuscript more suitable for a more technical journal and I do not recommend for the publication in *Nature Communications*.

Point-by-point response to the reviewers

Reviewer #1:

The manuscript presents the application of the interferometric autocorrelation technique on a Free Electron Laser SASE pulse by picking one single mode of the pulse's spectrum and splitting it into two replicas with a grating type split and delay mirror. The two replicas were focussed into a position sensitive ion time of flight spectrometer and delay dependent singly charged xenon ions signal was measured. The results demonstrate the successful adaptation of this technique to partially coherent pulses provided one can generate or select one single mode.

General remark:

The authors present a convincing technique to control the relative phase of two replicas of a coherent pulses on the attosecond time scale. They have taken into accounts remarks on the previous manuscript version by adding experiment perspectives and discussed in more details the characteristics and limits of the technique in the additional material. Nevertheless, the four experimental perspectives are discussed for as many different FEL conditions. As an example the measurements have been done for a wavelength of 38 nm while the requirement for the xenon giant dipole resonance probing requires 13.5 nm wavelength. Moreover, the authors discuss the possibility of probing charge dynamics in biomolecules with single FEL mode through low bunch charge operating conditions at FLASH. This, as showed in the response to referees, corresponds to 7 nm wavelength pulses, how good does the split and delay works at this wavelength? We also understand that in this particular case the dispositive should be used as a "regular" pump-probe delay stage where one take into account the overlap between the two pulses envelopes with a very large delay range (-50 to 574 fs). Part of this is discussed in the supplementary material but I think it should be discussed in the manuscript as it would demonstrate unambiguously the versatility of the dispositive.

To conclude, the authors present an interesting method to control the delay between two FEL pulses, allowing a phase control at the attosecond time scale, moreover they responded to the first manuscript comments and provided the additional informations required but I think they should make a real case of the capabilities and limits of the technique not only in the supplementary material but in the manuscript too. Provided those changes, I would recommend this work to be published in Nature communication.

(#1.1) We followed the advice of the referee and added a detailed discussion on the versatility, capabilities and limits of our experimental scheme in the main text (lines 181—198).

Additional remarks:

Lines 166-170: This paragraph should appear before because it is one of the best argument of the technique for future applications. Also we would like a reference (there is one in the response to the referees.)

(#1.2) We agree with the referee that the revised order of the corresponding paragraphs significantly improves the line of argumentation. In addition we added the reference requested by the referee.

Lines 163-166: The site selective excitation requires wavelength in the nm range. This argument would benefit from a discussion on short wavelenght limits of the setup in the manuscript.

(#1.3) In the revised manuscript, we discuss the scalability of our setup towards short wavelengths in lines 181—192 of the main text.

Line 165: LCLS too has the capability to cover the C, N and O K-edges.

(#1.4) We thank the referee for this comment. In the spirit of XUV attosecond interferometry we have previously focused our discussion on the M-edges covered by FERMI and FLASH. However, in the revised manuscript we added a note on the capability of LCLS (lines 178—180) covering the K-edges of the corresponding elements, which nicely connects to the new paragraph on the limits of our experimental scheme.

Reviewer #3:

The manuscript reports experimental demonstration of generating two replica pulses with phase control between them from a SASE FEL source, at the FLASH FEL with wavelength of 38 nm. In the updated version, the split and delay unit, which plays an important role in the experiment, was added with detailed descriptions in supplementary material part. More discussions have been added about the potential applications with this developed technique. Overall, this is an interesting scheme, overcoming the disadvantages of the SASE pulses which have poor longitudinal coherence would expand SAE FEL applications in the cases where require accurate phase control.

I would recommend the publication of this manuscript in this journal.

Some remarks:

The scheme chose a very narrow bandwidth smaller than a single spectral spike which introduces large fluctuation on the intensity after the mono. For the low charge mode with only one or two spikes, the fluctuation problem is similar. The authors explained in the reply of the single-shot feature and data sorting analysis. This is fine for some experiments. I think it would be useful to point out this fluctuation issue in the main text and discuss briefly that the single-shot characterization feature would overcome this issue in data analysis.

(#3.1) We follow the advice of the referee and added in lines 131—137: 'It is important to note that the experiment is capable of single-shot data acquisition and single-shot characterization of relative-phase differences and pulse intensities, thus the 100% intensity fluctuations behind the monochromator are not a problem. It goes without saying that in linear spectroscopy data averaging performed over many pulses does not present any difficulties. As far as non-linear spectroscopy is concerned, the intensity dependence of the systems response is measured simultaneously by sorting the data according to FEL pulse intensity.'

Reviewer #4:

The paper proposes an experimental method to perform Michelson-type interferometry using special type of combo grating mirrors in XUV spectral range (wavelength 38 nm) with attosecond temporal precision (15.8 as). By itself, the paper is well write and the experimental data are clear and convincing. However, few innovative results are present inside the manuscript. Indeed the general idea to use a couple of gratings as split and delay mirror to perform Michelson-type interferometry is not new in EUV spectral range and has been already successfully proposed in a very similar publication by Gebert et al. New J. Phys. 16 07304 (2014).

(#4.1) We do not agree with referee in this point. Gebert et al. performed measurements at 160 nm, which is without doubt in the ultraviolet (UV) spectral range and not in the extreme ultraviolet (EUV or XUV). Of course, this work has been cited in our original manuscript. In the present contribution we clearly demonstrate the generation of phase-coherent pulse replicas with sub-cycle control in the EUV. Here, due to the ionizing radiation a new window of opportunities opens up, because local electronic structure and dynamics becomes accessible, i.e. controllable.

Moreover, the use of poor longitudinal coherence source like SASE–FEL source imposes the use of a monochromator to enhance the spectral purity of the radiation illuminating the splitting and delay unit, having strong penalty for the final photon flux at the sample plane, in the current experimental data about 80 nJ. The authors claim about the possibility to use single spike operation of SASE-FEL using low charge e-bunch. However, using different scheme of light generation, seed-FEL has already demonstrated the possibility to control the relative phase between two different pulses with better (3 as, K.Prince et al. Nature Photon. 10, 176-179 (2016)) or at least comparable performance (15 as, Gauthier, D. et al. Phys. Rev. Lett. 116, 024801 (2016)) with respect to mechanical based devices without restriction on photon outputs. This work would have benefited by a simple experimental data demonstrating with a non-linear process the capacities of the proposed unit. For those reasons, I considered this manuscript more suitable for a more technical journal and I do not recommend for the publication in Nature Communications.

(#4.2) We agree with Reviewer #4 that the previous version did not emphasize clearly enough that the advancement of the demonstrated phase control not only lies in the derived attosecond precision - which is indeed comparable to what has been achieved at seeded FELs and has been referenced by the authors - but also in the accessible delay range of our mechanical device.

It is important to note that (i) phase-shifters between undulator modules of the seeded FEL FERMI allow at maximum a delay of ~ 300 as [a], a technique used in the experiment by Prince et al. (ii) The ‘twin pulse seeding’ by Gauthier et al. was limited to a minimum delay of ~ 280 fs to avoid interference between the laser seeds [b]. In contrast, we can control the relative delay between two pulses from -50 fs up to +574 fs with attosecond precision continuously. It allows bridging a huge gap of 300 as – 280 fs covering three orders of magnitude in time- and corresponding length scales for studies of ultrafast phenomena.

We rectified this situation in the revised manuscript by adding this important discussion and by giving the additional references [a, b].

[a] B. Diviacco et al., Phase shifters for the FERMI@Elettra undulators, in Proc. IPAC 2011, 3278–3280

[b] E. Ferrari et al., Widely tunable two-colour seeded free-electron laser source for resonant-pump resonant-probe magnetic scattering, Nat. Commun. 7, 10343 (2016)

DOI: 10.1038/ncomms10343

REVIEWERS' COMMENTS:

Reviewer #1 (Remarks to the Author):

The authors have added a more detailed discussion on the capabilities and limits of the technique which was missing in the previous version and they have taken into account other suggestions and comments. Therefore I would recommend this work to be published in Nature communication.

Reviewer #4 (Remarks to the Author):

The authors improve the article inserting a clear dissertation on phase coherent control possible at FEL sources. I agree on their comments. I still not fully convinced that the article in the present form demonstrates the potentiality of the used technique. Indeed a large part of the article includes foreseen potential applications of a technical improvement - the interdigital split and delay unit - with respect to currently available letterature.